# Early Evolution in Cancer: A Mathematical Support for Pathological and Genomic Evidence in Clear Cell Renal Cell Carcinoma

**DOI:** 10.3390/cancers15245897

**Published:** 2023-12-18

**Authors:** Annick Laruelle, Claudia Manini, José I. López, André Rocha

**Affiliations:** 1Department of Economic Analysis, University of the Basque Country (UPV/EHU), 48015 Bilbao, Spain; 2IKERBASQUE, Basque Foundation for Science, 48011 Bilbao, Spain; 3Department of Pathology, San Giovanni Bosco Hospital, ASL Città di Torino, 10154 Turin, Italy; claudia.manini@aslcittaditorino.it; 4Department of Sciences of Public Health and Pediatrics, University of Turin, 10124 Turin, Italy; 5Biomarkers in Cancer, Biocruces-Bizkaia Health Research Institute, 48903 Barakaldo, Spain; joseignacio.lopez@biocrucesbizkaia.org; 6Department of Industrial Engineering, Pontifical Catholic University of Rio de Janeiro, Rio de Janeiro CEP22451-900, Brazil; andre-rocha@puc-rio.br

**Keywords:** clear cell renal cell carcinoma, cancer evolution, game theory, intratumor heterogeneity

## Abstract

**Simple Summary:**

Clear cell renal cell carcinomas (CCRCCs) evolve as dynamic communities of individuals (cells) that are amenable to being studied under sociological rules. Here, the early period of development in CCRCC, progressing from initial homogeneity to high intratumor heterogeneity (ITH) and secondary clonal and sub-clonal diversification, is considered using the hawk-dove game. Fitness is a measure of biological aggressiveness in tumors. The results demonstrate that the fittest clone of a neoplasm in a heterogeneous context is fitter than the clone in a homogeneous environment in the early phases of tumor evolution. This study notes the advantages of a translational multidisciplinary approach in cancer research.

**Abstract:**

Clear cell renal cell carcinoma (CCRCC) is an aggressive form of cancer and a paradigmatic example of intratumor heterogeneity (ITH). The hawk-dove game is a mathematical tool designed to analyze competition in biological systems. Using this game, the study reported here analyzes the early phase of CCRCC development, comparing clonal fitness in homogeneous (linear evolutionary) and highly heterogeneous (branching evolutionary) models. Fitness in the analysis is a measure of tumor aggressiveness. The results show that the fittest clone in a heterogeneous environment is fitter than the clone in a homogeneous context in the early phases of tumor evolution. Early and late periods of tumor evolution in CCRCC are also compared. The study shows the convergence of mathematical, histological, and genomics studies with respect to clonal aggressiveness in different periods of the natural history of CCRCC. Such convergence highlights the importance of multidisciplinary approaches for obtaining a better understanding of the intricacies of cancer.

## 1. Introduction

Renal cell carcinomas rank among the top ten most common neoplasms in Western countries [1] and are a hot topic in modern medicine due to their morphological and genomic variability, complex etiopathogenesis, and resistance to treatment. More specifically, clear cell renal cell carcinoma (CCRCC), an aggressive form of renal cancer accounting for more than 70% of cases [2], currently poses an oncological challenge with promising therapeutic alternatives [3]. CCRCC is a paradigm of intratumor heterogeneity (ITH) and a test bench for new therapies.

Tumor intricacies provide an opportunity to incorporate new scientific approaches. For example, the application of ecological principles to cancer research has led to the coining of the term eco-oncology [4] and to cancer being considered as a social dysfunction [5]. These approaches have brought significant advances in the knowledge of cancer dynamics. As a result, cancer has been found to show at least four different evolutionary pathways: linear, branching, punctuated, and neutral [6]. With the exception of the neutral model, evolution over time is governed by Darwinian principles, where driver mutations generate different clones that make such ITHs unique, unrepeatable tumor cell communities in every case. An exhaustive genomic analysis shows that branching and punctuated models predominate in CCRCC [7].

The assumption that a neoplasm is a huge community of cells interacting with one another enables game theory to be applied to cancer analysis [8]. Many examples of this interdisciplinary cooperation can be found in the literature. In particular, we have recently analyzed the clinical consequences of ITH in CCRCC using the hawk-dove game, showing that the math supports the clinical evidence [9]. Thus, mathematics, histopathology, and genomics come together to demonstrate that tumor aggressiveness is linked to low ITH in the late temporal periods of CCRCC [10].

In continuation with our previous study [10], we focus in this work specifically on the early periods of tumor evolution in CCRCC, providing a mathematical support to the histological and genomic evidence. Coupled with [10], the reader will obtain a complete overview of CCRCC dynamics supported by a mathematical perspective.

## 2. Materials and Methods

### 2.1. Clinical Context

Large-scale sequencing studies have recently begun to reveal the complexities of this neoplasm. Although most CCRCCs become clinically evident in adulthood (peaking in the 6th–7th decades of life), the first steps of this tumor seem to appear much earlier, in childhood or adolescence, when von Hippel-Lindau (*VHL*) gene mutation happens in one allele of a few hundred cells as a result of chromothripsis [11]. After a dormant period of decades, when the *VHL* gene of the other allele mutates in these patients, the tumor initiates its evolutive course. On that course, up to seven deterministic pathways of temporal and spatial evolution have recently been detected, most of them being either branching- or punctuated-type models that lead to attenuated and accelerated clinical evolution, respectively [7]. However, the same study reveals that a small group of cases displaying only *VHL* gene mutations corresponding to a linear-type evolutionary model show indolent behavior [7]. These genomic data prompted us to investigate whether a mathematical approach would support them.

### 2.2. The Hawk-Dove Game

Here, we focus on the interactions of cells in three different temporal scenarios of CCRCC. More precisely, we consider cell interactions between elements bearing three well-known genetic driver mutations in CCRCC, i.e., mutations in *VHL*, *PBRM1*, and *BAP-1* genes. *VHL* gene mutation is the common initial step in the vast majority of CCRCCs, and *PBRM1* and *BAP-1* gene mutations are paradigmatic genetic disorders of these tumors with non-aggressive and aggressive behaviors, respectively [12]. For the sake of simplicity, other driver mutations (*SETD2*, etc.) and all of the passenger mutations, which are quite frequent in CCRCC, are not taken into account.

The three temporal scenarios depicted in Figure 1 establish early and late eco-evolutionary periods occurring in many CCRCCs, which correspond roughly to linear, branching, and punctuated temporal models. The late period, i.e., how a CCRCC with high ITH (three cell types, branching model) evolves towards a neoplasm with low ITH (two cell types, punctuated model), has also recently been modeled using the hawk-dove game [9]. Here, however, we focus on modeling the early period, i.e., how homogeneous tumors (one cell type, linear model) evolve into CCRCCs with high ITH (three cell types, branching model).

The hawk-dove game analyzes competition in biological systems [13]. Here, the game models the bilateral interactions between cells. In each encounter, a cell can behave aggressively, like a hawk, or passively, like a dove, to acquire a resource v. If one cell is aggressive and its opponent is passive, the first obtains the resource and the second gets nothing. If both cells are aggressive, there is a fight and the winner gets the resource, while the loser bears a cost c > v. Assuming that they both have the same probability of winning, the expected fitness (“payoff” hereafter) of each cell is (v − c)/2. If both cells are passive, one withdraws and gets nothing, while the other takes the resource. Assuming that they both have the same probability of withdrawing, the expected payoff of each cell is v/2. These contingencies are summarized in the following payoff matrix:

Let α denote the probability of behaving as a hawk, so that a cell can choose a so-called pure strategy under which it behaves with certainty as either a hawk (α = 1) or a dove (α = 0). Instead, the cell can choose a mixed strategy (0 < α < 1) in which it has a probability α of behaving as a hawk and 1-α of behaving as a dove. In a bilateral encounter, the expected payoff of a cell depends on its own behavior and on that of its opponent. For example, in Table 1, if both cells were to play a mixed strategy α, their expected payoffs from such a bilateral encounter would be α^2^(v − c)/2 + α(1 − α)v + 0(1 − α)α + (1 − α)^2^v/2.

In any game, cells that adopt a strategy leading to a higher payoff show both higher fitness and a higher cell replicating ratio. Consequently, the proportion of cells in the population that adopt this advantageous strategy increases over time and may eventually lead to the extinction of cells that adopt other, less fit competing strategies.

The hawk-dove game is studied in three different populations, depending on the temporal scenarios of CCRCC. The population may be homogeneous (linear model), heterogeneous with two types (punctuated model), or heterogeneous with three types (branching model). The types have no intrinsic significance, in the sense that any pair of cells plays the same game (i.e., faces the same payoff matrix as described above). On the other hand, types are important because cells might choose different strategies, leading them to behave differently depending on what type of opponent cell they compete with.

Examples of how tumor cells modify their behavior depending on the environment have been reported previously [14]. When the population is homogeneous (linear model) cells only encounter a single type of cell and can just adopt a single behavior. When the population is heterogeneous, cells may adopt different behaviors when they encounter different types of cells: For instance, they may behave passively when they encounter a certain type of cell and aggressively when they encounter a different type. A cell’s payoff in an encounter depends on the behaviors adopted by the cells, and on the proportions of the different types of cells.

We compare the three models on the basis of the payoffs obtained when cells adopt the evolutionarily stable strategy (ESS). The ESS [13] shows the resilience of a given incumbent strategy already being adopted by a population of cells against any other invading strategy in the following sense: Given that mutation ratios are very low during the cell replicating process, consider a population where most members play an ESS while a small proportion of mutants choose a different strategy. In that situation, each mutant’s expected payoff is smaller than the expected payoff of a “normal” individual, so mutants are driven out from the population.

An example of the ESS concept applied to the classic hawk-dove game, modeling a homogeneous tumor with only one cell type following an evolutionary linear model, can be found in a benign tumor in which the entire population of cells behave passively and can be invaded by a mutant cell that adopts an aggressive behavior. From Table 1, if a single mutant cell invades the population adopting a hawk strategy, all but one of the incumbent cells will meet another passive cell, leading to an expected payoff v/2 for the incumbent cells in each encounter. However, the invading mutant cell will get a payoff v from meeting a passive cell, i.e., a payoff twice as high, giving the cell with the mutation the competitive advantage of replicating faster. In the next period, when cells compete again in such a population, the proportion of passive incumbent cells will have decreased, and that of aggressive mutant cells will have increased. Thus, the dove strategy is not an ESS in such a game, given that it can be invaded. As shown in [13], the ESS in the homogeneous game is the mixed strategy v/c: Cells play hawk with a probability (or frequency) of v/c and play dove with a probability of 1 − v/c. The larger the resource, the more often cells act aggressively, and the larger the cost the more often cells act passively.

In [15], the ESS in heterogeneous populations with two types are studied. The ESS depends on the proportions of the different cells. Some discrimination also arises: One type is always treated better than the other.

Laruelle et al. [9] demonstrate that in heterogeneous populations with three types, there is (under some conditions) an ESS in which each type of cell receives a different treatment.

The comparison of the payoffs in the ESS of the different population games yields the following results (see [9,13,15] and Appendix A for the proofs).

## 3. Results

Result 1: When the population is heterogeneous (i.e., in the punctuated and branching models) one type of cell (the most malignant cells, hereafter “A-cells”) obtains a strictly larger payoff than the other in the ESS.

Result 1 suggests that a heterogeneous environment favors discrimination. Cells face the same payoff matrix in all encounters, but the possibility of differentiating cells generates different behaviors when different cells are faced.

Result 2: The payoff that an A-cell obtains in the ESS in a heterogeneous population is strictly larger than that which it would obtain in the ESS in a homogeneous population.

Result 2 suggests that there is an advantage for A-cells in being in a heterogeneous environment rather than a homogeneous one. The intuition is that in homogeneous environments, all cells are treated equally and receive the same expected payoff. In a heterogeneous environment, discrimination arises: Some cells (the A-cells) are treated better than they would be in the homogeneous environment, and others are treated worse. In consequence, A-cells receive a larger payoff than they would in a homogeneous environment.

Result 3: For a given proportion of A-cells, the payoff that an A-cell obtains in the ESS in a punctuated model is strictly larger than it would obtain in the ESS in a branching model (if there is an evolutionarily stable strategy).

Result 3 suggests that there is an advantage for some cells in being in a punctuated environment rather than a branching environment. That is, whenever there is discrimination, it is better for A-cells when there are two groups of cells than when there are three.

## 4. Discussion

Cancer is a disease with a high impact in Western countries, whose complexities defy human understanding. Cell-to-cell interactions govern the temporal and spatial evolution of the disease, which means that every case is unique and unrepeatable. However, certain deterministic evolutionary routes enable future behavioral events to be predicted. Figure 1 illustrates the natural evolution of many CCRCCs.

Mitchell et al. [11] have recently revealed the first steps in the carcinogenesis of these neoplasms, identifying long periods of dormancy prior to the symptomatic phase of the neoplasm. At least theoretically, the initial stages of tumor chronology generate a homogeneous cell growth driven by a *VHL* gene mutation. A few of these tumors will pursue a linear evolution throughout their natural history, the so-called *VHL*-driven CCRCCs, but most will develop subsequent driver mutations, thus generating distinct clones within the tumor [7]. Such new clones (and sub-clones) evolve in a Darwinian model of coexistence, are responsible for ITH, and determine tumor evolution and patient prognosis. VHL syndrome-associated CCRCCs are characteristically multiple and bilateral low-grade neoplasms in which inter-tumor heterogeneity takes place along the natural history. Many CCRCCs are surgically removed from patients during the branching-type period because they usually become symptomatic at that time. Other cases, however, evolve quicker and farther due to the intrinsic aggressiveness of one of these newly generated clones. *BAP-1* gene mutation is a paradigm of these high-fitness clones. Again, following a Darwinian evolutionary pattern, this highly aggressive clone installs the punctuated period in the tumor, dominates subsequent tumor overgrowth, and prompts metastatic development due to local hypoxic pressures in the tumor interior [16].

The later evolutionary step from branching to punctuated model is illustrated in Figure 1. It carries a transition from high to low ITH. Interestingly, the clinical evidence shows that aggressive forms of CCRCC display low ITH at the histological level. Manini et al. [17] have performed a thorough histological analysis of a number of CCRCCs and found that patients with tumors with low variability in the histological grade across the sampling died of the tumor or were symptomatic at last clinical contact, while patients with tumors with a broader spectrum of histological grades were all alive and without clinical disease. Other authors have found that the metastatic ability of a tumor is not always strictly linked to the clone with the highest histological grade [18], so taking grade variations, and not only the highest grade, into consideration may matter in the pathologist’s work-up. This simple idea may have promising practical applications for pathologists. Interestingly, genomic analyses also point to the same conclusion. An exhaustive study of more than 1200 tumor regions in 101 CCRCCs has shown that tumors with high chromosomal complexity and low ITH pursue an aggressive clinical course, with early and multiple metastases, while those with low chromosomal complexity and high ITH behave in an attenuated fashion with late, solitary metastases [7]. The conclusion at this point of tumor evolution is that high ITH is linked to an attenuated evolutive course and longer survival than with low ITH tumors. This evidence is supported by a recent mathematical approach using the hawk-dove game [9].

Therapy influences ITH evolution, and this issue deserves comment. Current opinion advocates using the maximum tolerable doses as the preferred strategy, but this forces tumor cells to develop resistances. Antiangiogenic drugs, for example, block neo-angiogenesis in CCRCC, prompting tumor necrosis. However, some tumor cells attempt to generate resistance to such drugs, making it possible for a new tumor made up only of resistant cells to develop. The final result is that maximum tolerable doses turn an initial high ITH (branching-type tumor) into a low ITH (punctuated-type tumor), which is much more aggressive. Promising alternative therapy strategies based on mathematical analyses in the form of the so-called “adaptive therapies” [19] propose the implementation of ecological principles to manage cancer and suggest that the preservation of a high ITH may promote intercellular competition, which could slow down tumor progression. In normal conditions, tumor cells expend their energy on increasing their fitness in an intercellular, competitive way to accomplish all cellular functions. Since energy is limited, a therapeutic strategy using drugs below the maximum tolerable doses will force tumor cells to diversify their energy expenditure across several functions, e.g., increasing fitness and generating resistance, thus slowing down both actions. Interestingly, this double effect means increased survival for patients.

In this paper, however, we focus on the early evolutionary steps of CCRCC from linear to branching model, bringing the transition from homogeneous to high ITH tumors. Our mathematical analysis concludes that high ITH CCRCCs (three cell-type, branching model) are more aggressive than their homogeneous counterparts (one cell-type, linear model). This is the conclusion obtained in the hawk-dove game, and histopathological/genomic studies support it. For example, seven out of 101 CCRCCs in the multi-region genomic analysis performed by Turajlic et al. [7] were made of homogeneous tumor cell populations bearing only *VHL* gene mutations. After the clinical follow-up, this specific subset of tumors behaved indolently and never metastasized [7]. Similarly, the histological analysis of 28 totally sampled CCRCCs by Manini et al. [17] revealed that three of them were totally homogeneous low-grade neoplasms (Grade 1), being alive and without disease after a long-term follow-up. Whether these indolent CCRCCs had evolved over time towards aggressive forms of the disease via subsequent potential clonal/subclonal diversification remains debatable, but possible, and highlights the crucial importance of the exact time of removal in our knowledge of tumor characteristics. A recent review of the importance of ITH in breast, lung, hepatic, and colorectal cancer agrees with this statement [20,21,22,23,24,25,26,27].

The apparent contradiction in the aggressiveness of high ITH is resolved when it is taken into account that cancer is a dynamic disease in which pathologists analyze only static snapshots obtained at the time of tumor removal. That time is variable and not programmable, and depends on multiple clinical factors. A high ITH can thus be better or worse depending on the exact period of tumor evolution and analysis. The natural history of every tumor is not evident in routine practice, and regrettably, current therapies do not factor the evolutionary pattern of tumors into their strategies.

In recent years, game theory has been gaining momentum in oncology. This study notes the importance of translational research in modern medicine and highlights once again the importance of mathematical analyses as an allied tool in cancer analysis [10].

## 5. Conclusions

Here, we consider the initial period in the evolution of many CCRCCs, specifically the time when tumors progress from the original cellular homogeneity towards high ITH as a consequence of clonal and sub-clonal development. Four points derived from our analysis deserve mention: (i) cancer is a dynamic disease in which pathologists analyze a static moment of its evolution; this time is variable and not programmable, (ii) tumors evolve throughout their natural history and may display different degrees of ITH at different times, (iii) this study focuses specifically on the early moments of CCRCC evolution, which is not always paired to clinical findings and diagnosis of the disease, and (iv) the natural history of every tumor is not evident in routine practice and, therefore, not exploitable.

Once more, mathematics supports the clinical evidence and needs to be considered as a tool in the armamentarium for better understanding cancer complexities.

## Figures and Tables

**Figure 1 cancers-15-05897-f001:**
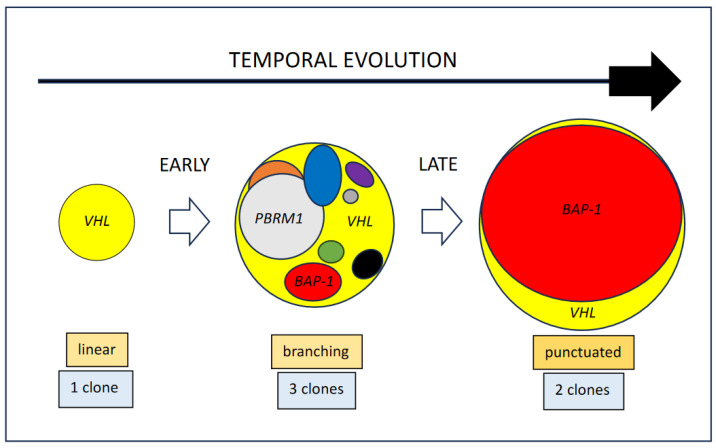
Example of a model of temporal evolution in clear cell renal cell carcinomas. The size of the circles reflects tumor aggressiveness (level of clonal fitness). Early evolution exemplifies the transition from linear (homogeneity (*VHL* monodriver clone)) to branching (high heterogeneity (*VHL*, *PBRM1*, and *BAP-1* driven clones)) models. However, late evolution shows the transition from branching (high heterogeneity (*VHL*, *PBRM1*, and *BAP-1* driven clones)) to punctuated (low heterogeneity (*VHL*, *BAP-1* driven clones)) models due to the expansion of an aggressive clone.

**Table 1 cancers-15-05897-t001:** Hawk-dove game payoff matrix.

	Hawk	Dove
Hawk	(v − c)/2	v
Dove	0	v/2

## Data Availability

The data presented in this study are available in this article.

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
