# Peer review of "Early Evolution in Cancer: A Mathematical Support for Pathological and Genomic Evidence in Clear Cell Renal Cell Carcinoma"

_cancers, 2023, doi:10.3390/cancers15245897_

Round 1

Reviewer 1 Report

Comments and Suggestions for Authors

This article appears to be a complex scientific paper focused on the evolution of renal cell carcinoma and the role of intratumor heterogeneity (ITH) in its development. While the content is undoubtedly scientific and informative, there are several aspects that could be improved:

  • Clarity and Readability:
    • The article is quite technical and contains multiple complex terms and concepts. It could benefit from clearer and more concise language to make it accessible to a broader audience.
    • The organization of the content is somewhat challenging to follow. Consider structuring the article with clear subheadings and bullet points to break down complex ideas.
  • Introduction:
    • The introduction should provide more context and motivation for the study. Why is renal cell carcinoma important, and what is the current state of research in this field?
    • Include a clear research question or hypothesis that the study aims to address.
  • Acronyms and Abbreviations:
    • Define acronyms and abbreviations upon their first use. For example, CCRCC, ITH, ESS, and VHL should be explained when they first appear.
  • Methods Section:
    • The "Materials and Methods" section is incomplete. It mentions the Hawk-Dove game and genetic driver mutations, but it lacks detailed information about the methods used in the study. Provide a clear description of the methodology.
  • Discussion Section:
    • The discussion section is quite lengthy and somewhat disorganized. Consider breaking it down into subsections to address specific points.
    • It would be beneficial to summarize the main findings and their implications more clearly.
    • When talking about tumor heterogeneity you should take into consideration to stress also genetic disease as Von Hippel–Lindau syndrome. This is a rare genetic disorder with multisystem involvement. About 70% present RCC and among those multiple and bilateral lesions are possible. Each of these lesions presents a high level of ITH. It would be interesting to better describe this concept. Please include and discuss also the following: PMID: 37685983; PMID: 37373581; 
  • Conclusions:
    • The conclusion is somewhat abrupt. It should restate the key findings and their significance in the context of the broader field of research.

In summary, the article contains valuable scientific information but requires improved clarity, structure, and presentation to make it more accessible and understandable to a broader audience.

Author Response

This article appears to be a complex scientific paper focused on the evolution of renal cell carcinoma and the role of intratumor heterogeneity (ITH) in its development. While the content is undoubtedly scientific and informative, there are several aspects that could be improved:

We thank very sincerely the efforts and comments of the reviewer. Yes, the paper puts together two very distant areas of science and this fact makes both writing and understanding a bit more difficult compared with other more conventional papers. Thus, papers “frontier” in any area are always a challenge. In this case, the application of a theoretical mathematical approach to a practical issue like the analysis of cancer evolution leads to some grey zones very difficult, if not impossible, to be well explained by pathological tools, since pathology offers basically a snapshot of a dynamic process. We will try to follow the suggestions to clarify some issues.

Clarity and Readability:

  • The article is quite technical and contains multiple complex terms and concepts. It could benefit from clearer and more concise language to make it accessible to a broader audience.

The language used includes terms used in genomics (clones, subclones, driver mutations, passenger mutations chromothripsis, etc.) of common use in cancer research. We think that it is out of the scope of the paper to define them. The concept of intratumor heterogeneity reflects the dynamic evolution that all cancers follow during their natural life and types of cancer evolution (linear, branching, punctuated, neutral), although of recent introduction, are well known and accepted in oncology. Anyway, the reference #6 (Davis et al, 2017) in the bibliography refers exactly to that, and the reader needing additional information can go there for a precise definition. Mathematical terminology has been reduced at maximum, less is impossible unless information is lost, and most of the difficulties appear in the appendices, which are not necessary for a general understanding of the paper. Nonetheless, although difficult to understand outside mathematics, these appendices are needed to scientifically support with robustness our mathematical statements. Anyway, we have included additional explanations connecting with oncology in the last paragraphs of the discussion (in red).

  • The organization of the content is somewhat challenging to follow. Consider structuring the article with clear subheadings and bullet points to break down complex ideas.

We honestly believe that the organization of the paper is standard and clearly identified. Here, we list the subheadings we have used:

Simple Summary

Abstract

  1. Introduction
  2. Materials and Methods

2.1 Clinical Context

2.2 The Hawk-Dove game

  1. Results
  2. Discussion
  3. Conclusions
  4. References

Introduction:

  • The introduction should provide more context and motivation for the study. Why is renal cell carcinoma important, and what is the current state of research in this field?

The introduction contextualize and explains the motivation of the study and has been organized as follows:

  1. Lines 37 to 43 mention why renal cell carcinoma is important in terms of frequency, clinical aggressiveness, resistance to therapies.
  2. The second paragraph (lines 44 to 53) introduces the first notions of how ecological principles may help in the understanding of cancer and mention the four accepted evolutionary patterns and the Darwinian/non-Darwinian concept in tumor dynamics.
  3. The third paragraph (lines 54 to 61) links cancer, particularly in this case clear cell renal cell carcinoma, with Game Theory. This new approach is gaining momentum in the last years providing clues to understand, and predict, cancer behavior.
  • Include a clear research question or hypothesis that the study aims to address.

Yes, for such a purpose, we have removed the last sentence of the last paragraph of the discussion and have rephrased the idea in a short new paragraph to finish the Introduction chapter. The sentence reads as follows: In continuation with our previous study [10], we focus in this work specifically on the early periods of tumor evolution in CCRCC providing a mathematical support to the histological and genomic evidence. Coupled with [10] the reader will obtain a complete overview of CCRCC dynamics”. (in red).   

  •  Acronyms and Abbreviations:
    • Define acronyms and abbreviations upon their first use. For example, CCRCC, ITH, ESS, and VHL should be explained when they first appear.

Yes, the correct use of acronyms have been changed accordingly (in red).

  •  Methods Section:
    • The "Materials and Methods" section is incomplete. It mentions the Hawk-Dove game and genetic driver mutations, but it lacks detailed information about the methods used in the study. Provide a clear description of the methodology.

The methodology is to apply the hawk-dove game to the evolution (analyzed by histological and genomic perspectives) of CCRCC. The hawk-dove game is a game taking part of the Game Theory which analyzes, as explained in the text, interactions between individuals in biological contexts. It is widely used to analyze cancer cell behaviors. It is supported and explained in detail in references #13 and #15. Genetic driver mutations in CCRCC are responsible of intratumor heterogeneity in many carcinomas, CCRCC included. Such driver mutations have been detailed and supported in references #7, #11, and #16. The goal of the paper is not the description of any type of mutations.

  •  Discussion Section:
    • The discussion section is quite lengthy and somewhat disorganized. Consider breaking it down into subsections to address specific points.

We consider the discussion follows a logical course.

The first paragraph mentions the complexities of cell-to-cell interactions in CCRCC and present the figure in which the evolution is depicted in a schematic form.

The second paragraph deep into the molecular mechanisms underlying such complexity and introduces the evolutionary concepts governing cancer.

The third paragraph goes directly to the histological and genomic evidence with which mathematics will confront.

The fourth paragraph connects intratumor heterogeneity with therapeutic resistances and consider that ITH is responsible of many cancer deaths, and propose that a different approach to treatment, based on ecological principles but supported by robust mathematical analyses, may help patients and improve their survival rates.

The fifth paragraph connects the hawk-dove game with intratumor heterogeneity in the late periods of tumor evolution.

The two last paragraphs try to justify an apparent contradiction considering that any tumor show different stages across their evolution and stress the importance of the time of tumor removal.  

    • It would be beneficial to summarize the main findings and their implications more clearly.

Yes, we have included the three important points detected in this approach. For such a purpose, taking also the suggestion of the second reviewer, the following points has been included in the discussion:

  1. cancer is a dynamic disease, in which pathologists analyze a static moment of its evolution. This time is variable and not programmable.
  2. tumors evolve dynamically throughout their natural history and may display different degrees of ITH in different times.
  3. this study focuses on the early moments of tumor evolution, which not always is paired to clinical findings and diagnosis of the disease.
  4. the natural history of every tumor is not evident in routine practice and therefore not exploitable.

    • When talking about tumor heterogeneity you should take into consideration to stress also genetic disease as Von Hippel–Lindau syndrome. This is a rare genetic disorder with multisystem involvement. About 70% present RCC and among those multiple and bilateral lesions are possible. Each of these lesions presents a high level of ITH. It would be interesting to better describe this concept. Please include and discuss also the following: PMID: 37685983; PMID: 37373581; 

A mention to CCRCC associated to VHL syndrome has been included in the discussion (in red). The papers proposed to be included do not matter with the topic. In fact, PMID 37685983 deals with cancer stem cells in renal cell carcinoma and PMID 37373581 deals with tumor microenvironment. As they do not add anything specific to the goal of our manuscript we have not included them.

  • Conclusions:
    • The conclusion is somewhat abrupt. It should restate the key findings and their significance in the context of the broader field of research.

We have included four main conclusions making the conclusion more readable.

In summary, the article contains valuable scientific information but requires improved clarity, structure, and presentation to make it more accessible and understandable to a broader audience.

Reviewer 2 Report

Comments and Suggestions for Authors

The authors should be congratulated for the work.

The aim of this study is to exploit for future therapeutical options the convergence of mathematical, histological, and genomics studies with respect to clonal aggressiveness in different periods of the natural history of CCRCC. Such union and team work highlights the role and importance of multidisciplinary approaches for obtaining a better understanding of the intricacies of cancer.

The results are very interesting and suggest how a future therapeutical options and approach may consist in implementation of ecological principles to try and manage cancer, taking advantage of the fact that the preservation of a high ITH may promote intercellular competition, which could slow down or maybe even arrest tumor progression, increasing patient survival.

The Darwinian concept of evolution and survival applied to tumoral behavior is very original and interesting. 

Of course, this study has its limits, as clearly announced by the authors:

- cancer is a dynamic disease, in which pathologists analyze a static moment of the tumor's behavior in that specific moment. This time is variable and not programmable.

- tumors evolve dynamically throughout their natural history and may display different degrees of ITH in differente moments of time.

- this study focuses on initial moments of tumor genesis and evolution, which not always is relatable to clinical findings and diagnosis of the disease.

- the natural history of every tumor is not evident in routine practice and therefore not exploitable.

Comments on the Quality of English Language

The English language used appears to be quite correct and clear.

Minor revision of the language may be considered.

Author Response

The authors should be congratulated for the work.

The aim of this study is to exploit for future therapeutical options the convergence of mathematical, histological, and genomics studies with respect to clonal aggressiveness in different periods of the natural history of CCRCC. Such union and team work highlights the role and importance of multidisciplinary approaches for obtaining a better understanding of the intricacies of cancer.

The results are very interesting and suggest how a future therapeutical options and approach may consist in implementation of ecological principles to try and manage cancer, taking advantage of the fact that the preservation of a high ITH may promote intercellular competition, which could slow down or maybe even arrest tumor progression, increasing patient survival.

The Darwinian concept of evolution and survival applied to tumoral behavior is very original and interesting. 

Of course, this study has its limits, as clearly announced by the authors:

- cancer is a dynamic disease, in which pathologists analyze a static moment of the tumor's behavior in that specific moment. This time is variable and not programmable.

- tumors evolve dynamically throughout their natural history and may display different degrees of ITH in differente moments of time.

- this study focuses on initial moments of tumor genesis and evolution, which not always is relatable to clinical findings and diagnosis of the disease.

- the natural history of every tumor is not evident in routine practice and therefore not exploitable.

Thank you for your comments. We have included your four last point in the conclusions since they reflect our limitation approaching cancer as a dynamic disease.

Round 2

Reviewer 1 Report

Comments and Suggestions for Authors

The manuscript has been improved in almost all the highlighted aspects, but I will defer the decision to the editor